# Segmentation of the ECG Signal by Means of a Linear Regression Algorithm

**DOI:** 10.3390/s19040775

**Published:** 2019-02-14

**Authors:** Javier Aspuru, Alberto Ochoa-Brust, Ramón A. Félix, Walter Mata-López, Luis J. Mena, Rodolfo Ostos, Rafael Martínez-Peláez

**Affiliations:** 1Faculty of Mechanical and Electrical Engineering, University of Colima, Av. Universidad #333, Colima 28000, Mexico; jaspuru@ucol.mx (J.A.); rfelix@ucol.mx (R.A.F.); wmata@ucol.mx (W.M.-L.); 2Academic Unit of Computing, Master Program in Applied Sciences, Polytechnic University of Sinaloa, Mazatlan 82199, Mexico; lmena@upsin.edu.mx (L.J.M.); rostos@upsin.edu.mx (R.O.); 3Faculty of Information Technology, University of La Salle-Bajio, Av. Universidad #602, Leon 37150, Guanajuato, Mexico; rmartinezp@delasalle.edu.mx

**Keywords:** segmentation, Digital Signal Processing, ECG Sensor, Linear Regression Algorithm, identification waves

## Abstract

The monitoring and processing of electrocardiogram (ECG) beats have been actively studied in recent years: new lines of research have even been developed to analyze ECG signals using mobile devices. Considering these trends, we proposed a simple and low computing cost algorithm to process and analyze an ECG signal. Our approach is based on the use of linear regression to segment the signal, with the goal of detecting the R point of the ECG wave and later, to separate the signal in periods for detecting P, Q, S, and T peaks. After pre-processing of ECG signal to reduce the noise, the algorithm was able to efficiently detect fiducial points, information that is transcendental for diagnosis of heart conditions using machine learning classifiers. When tested on 260 ECG records, the detection approach performed with a Sensitivity of 97.5% for Q-point and 100% for the rest of ECG peaks. Finally, we validated the robustness of our algorithm by developing an ECG sensor to register and transmit the acquired signals to a mobile device in real time.

## 1. Introduction

The electrocardiogram (ECG) signal reflects the electrical activity of the heart observed from the strategic points of the human body and represented by quasi-periodic voltage signal. The ECG signal contains essential information about the cardiac pathologies affecting the heart, characterized by five peaks known as fiducial points, which are represented by the letters P, Q, R, S, and T (Figure 1) [1,2]. The QRS complex is the depolarization of the right and left heart ventricles, which is used as a reference point for signal analysis. The P wave is the result of the depolarization of the atrium, while the ventricle causes the rest of the peaks. The diagnosis of the signal relies on the morphology of the waves, as well as the duration of each peak and the segments that make it up. Therefore, detection of each section of the ECG signal is essential for health professionals in screening, diagnosis, and monitoring of several heart conditions [3,4].

The detection of the QRS complex of the ECG signal was assessed in [5,6]. From a medical point of view, essential information present in the ECG signal are included in the P wave, the QRS complex, and the T wave. These data include the duration of the PR and QT intervals, and the PR and ST segments. However, detection and discrimination among ECG sections become complicated due to the variable physiology of the reference points; mainly produced by cardiac abnormalities, alterations of the isoelectric line or noise added to the ECG signal. Therefore, to improve diagnosis of heart diseases from ECG information, all these corruption problems must be addressed.

Recently, several methods have been proposed to improve the detection of ECG waves. Among these methods are the Pan-Tompkins algorithm [7], the Wavelet transform (WT) using a constant scale in signal analysis, without considering the characteristics of the signal [8,9], and machine learning approach using artificial neural networks, which, through controlled training, analyze ECG signals pre-diagnosed and separate them into known classes, identifying new signals, and therefore their diagnosis [3]. 

In this sense, a novel technique for extracting the non-linear and non-stationary characteristics of ECG signals is the Hilbert-Huang transform mixed with the WT, through which it is possible to detect efficiently peaks in the ECG beat even with the presence of a low signal-to-noise ratio [9]. In this order, the Hermite transform has also been proposed as an optimized approach for the compression and clustering of QRS complexes [10,11].

On the other hand, a method based on Shannon energy (SE) envelope to detect QRS complex in 12 leads of ECG signal was provided in [12]. This approach estimates the points where the highest concentration of average energy in the spectrum, for better detecting peaks in case of various QRS polarities and sudden changes in their amplitude. Initially, a band-pass filter is used for eliminating noise and SE of ECG signal is calculated. Later, an envelope of SE is used to define a specific threshold and detect peaks.

Furthermore, Yeh & Wang [13] proposed a simple and reliable method termed the Difference Operation Method to detect the QRS complex through two steps. The first is to find the R point by applying the difference equation operation to the ECG signal. In the second stage, the Q and S points are searched using the R point as a reference to find the QRS complex. 

Conversely, Akhbari et al. used a Multi Hidden Markov model (HMM) for extracting fiducial points of ECG signals [14]. In the multi HMM approach, each segment of an ECG beat is represented by a separate ergodic continuous density HMM. Each HMM is trained separately to compare the log-likelihood of two consecutive HMMs and estimate a path, which shows the correspondence of each part of the ECG signal to the HMM with the maximum log-likelihood.

In consequence, most techniques for the detection of the P and T peaks require the QRS complex as a reference point. The purpose of segmenting the ECG signal is to locate the waves, segments, and intervals and carry out the comparison of these with known patterns, through their characteristics of time and morphology. This process can be done manually or automatically by designing classification systems that can detect possible pathologies present in the ECG signal. However, to develop classification systems more accurate, it is necessary to extract the most relevant information present in the ECG signal [15].

In this study, we propose a novel method for detecting fiducial points of ECG waves, using linear regression to identify maximums and minimums from an acquired ECG signal. The first stage consisted of pre-processing the signal using a low pass filter to reduce the noise present in the signal [16]. Then, we calculated a low-order polynomial function, which was the approximation of the deviation of the signal as its amplitude varies from the baseline. When we subtracted the values of the polynomial function from the amplitude of the ECG signal, it returned the isoelectric level of the signal [17]. The second stage consisted of detecting the R point and the value of the isoelectric line, already linearized, through the proposed algorithm. The ECG signal was then segmented into periods, using the average distance among all R points as window size. The third stage consisted of applying the algorithm again throughout the ECG signal to detect the maximum and minimum points within an analysis window, which generated information on the location of the P, Q, R, S, and T peaks. 

We performed the mathematical analysis using the MATLAB software [18] (mathematical software tool), which allowed to obtain the coordinates of the maximum and minimum points corresponding to the location of the ECG peaks. The signals used in this paper are part of the "ECG-ID" database of the PhysioNet biological signal bank [19,20]. 

Finally, we tested the detection algorithm with signals acquired in real time by an ECG sensor that included data acquisition, amplification, filtering, digitization, and transmission from ECG signals analyzed by the proposed method. In Figure 2 we show the general architecture of the process carried out in physical tests, in which the ECG sensor records the signal at a low level to generate the ECG signal that a microcontroller unit (MCU) with a Bluetooth Low Energy (BLE) module will digitize and transmit wirelessly for further processing. The signal is received and processed by the detection approach, which provides the identification of P, Q, R, S, and T peaks.

## 2. ECG Signal Processing

The analog-to-digital conversion process of ECG signals can cause different kinds of noise and interference that affect the quality of the information contained in the signal [21]. The causes of noise can be various: power-line interference (with frequency 50 Hz or 60 Hz), deviation from the baseline wander, or electrical activity of the muscles (EMG). In the ECG signal, EMG interference appears as rapid fluctuations that vary faster than ECG waves, and their operating frequencies are in the range of 0.01 Hz to 10 kHz [14].

The frequency components of a typical ECG signal are in the range of 0.05–100 Hz [2]; therefore, most approaches use filters to eliminate unwanted signals that are outside this range. The application of a Butterworth low-pass filter provides a smooth transition between the frequencies that belong to passband and the frequencies that are over cut band. This allows to maintain essential information after of eliminating high frequencies present in an ECG signal. 

The mathematical formulation of Butterworth filters is written as follows:(1)$$H(f)=\frac{B(f)}{A(f)}=\frac{b_{1}{\left(jf\right)}^{n}+b_{2}{\left(jf\right)}^{n-1}+\cdots +b_{n+1}}{a_{1}{\left(jf\right)}^{n}+a_{2}{\left(jf\right)}^{n-1}+\cdots +a_{n+1}}$$
where *j* is the imaginary unit, and the coefficients *a_i_* and *b_i_*, are the filter coefficients. The order *n* identifies both the shape and complexity of the filter, hence a higher *n* provides a sharper filter; however, this will be more complex [16]. The ECG signal was filtered using MATLAB with a Butterworth filter of order 10 at a cut-off frequency of 100 Hz (Figure 3).

After filtering, it is necessary to eliminate the baseline wander, since the data in the ECG signal do not represent the real values of the amplitude. To eliminate this deviation, a low-order polynomial function was coupled to the signal and used to align the isoelectric line. The polynomial coefficients *P(x)* of order *N* were obtained by applying the MATLAB tool, to obtain the polynomial function that best matched the ECG signal data in the least squared sense. In this sense, a polynomial function of order *N* = 8 was used, and with the *polyfit* command we built a vector of length *N* + 1, which contained the coefficients of the polynomial in descending order; while with the *polyval* command, we obtained the values of the polynomial *P* evaluated for each point of the *x*-axis. This vector was then subtracted from the ECG signal, with the goal of reducing to a minimum its amplitude with respect to the isoelectric line [17]. Figure 4 shows two ECG signals; in the first, the signal presented a variation along the *x*-axis, while in the second, the baseline wander was reduced after the subtraction of values of the polynomial function. 

## 3. Detection of *R*-Point and Isoelectric Line

Once the filtered signal y(n) was obtained, the next step was to find the peaks of the *R* wave and the amplitude value of the isoelectric line.

### 3.1. Peak Detection Algorithm

Let y(n) be a given ECG sequence for n∈{0,1,2,…,N−1}, where *N* is the number of samples of the signal. The quadratic mean parabola-fit error ε(i) is defined as:(2)$$\varepsilon (i)=\frac{1}{2w+1}\sum_{j=i-w}^{i+w}{\left(y(j)-y^{\prime }(j)\right)}^{2}$$

ε(i) is a sequence for i∈{w,w+1,w+2,⋯,N−w−1}, where w is a positive number such that, 2w+1<N. On the other hand, y(j) is a sequence, fragment of y(n), centered at index, therefore, j∈{i−w,i−w+1,i−w+2,⋯,i,i+1,i+2,⋯,i+w−1,i+w}.

The length of both sequences y(j) and $y^{\prime }(j)$ is m=2w+1. The sequence $y^{\prime }(j)$ is defined by a parabola, whose vertex is at index *i*, that best fit the ECG signal fragment y(j), where *j* is defined by Equation (4). Hence, $y^{\prime }(j)$ can be written as:(3)$$y^{\prime }(j)=a{\left(i-j\right)}^{2}+v$$

For sake of clarity, in Figure 5 are plotted an example of sequences y(n), y(j), and $y^{\prime }(j)$.

Calculating the values for a and v that make $y^{\prime }(j)$ to best fits y(j) is done by the well-known approach of linear regression. At first, the quadratic function of Equation (3) is changed to a linear one, being that the indexes *j* and *i* are known, variable $x={\left(i-j\right)}^{2}$ replaces the whole set.
(4)$$y^{\prime }(x)=ax+v$$

The vertex of the parabola will be located at (i,v); it is necessary to find the value v of the vertex and the opening coefficient a that best fits $y^{\prime }(j)$ to y(j). Finding these points is possible using the regression line [21], as it is feasible to treat Equation (4) as a line. We calculated these values with Equations (5) and (6):(5)$$a=\frac{m\sum_{j=l_{1}}^{l_{2}}x(j)y(j)-\sum_{j=l_{1}}^{l_{2}}x(j)\sum_{j=l_{1}}^{l_{2}}y(j)}{m\sum_{j=l_{1}}^{l_{2}}x^{2}(j)-{\left(\sum_{j=l_{1}}^{l_{2}}x(j)\right)}^{2}}$$
(6)$$v=\frac{\sum_{j=l_{1}}^{l_{2}}x^{2}(j)\sum_{j=l_{1}}^{l_{2}}y(j)-\sum_{j=l_{1}}^{l_{2}}x(j)y(j)\sum_{j=l_{1}}^{l_{2}}x(j)}{m\sum_{j=l_{1}}^{l_{2}}x^{2}(j)-{\left(\sum_{j=l_{1}}^{l_{2}}x(j)\right)}^{2}}$$
where *m* is the window size m=2w+1, and the limits *l*_1_ and *l*_2_ are defined as:(7)$$\begin{array}{l} l_{1}=i-w\\ l_{2}=i+w \end{array}$$

In this way, it is possible to generate a parabola $y^{\prime }(j)$ that can adjust to the values contained in the time window, which will go through the entire signal. When there is a peak in the processed signal y(i), the vertex of the parabola (i,v) will coincide with the maximum or minimum point of the wave. It is possible to detect these points in the waves comprising the signal y(i), using the error signal generated by Equation (2) and comparing the signal y(j) to the parabola $y^{\prime }(j)$.

The window size for *R* point detection is defined by the average duration of the QRS complex, which is from 70 to 100 ms [15], taking only the approximate width of the *R* peak and being *f_s_* equal to the frequency with which the signal was sampled, the number of samples of the half of the window will be equal to:(8)$$w=0.030\times f_{s}$$

The signal ε(i) will be fixed along where the quadratic approximation of the signal y(n) was made and at smaller values of ε(i), the quadratic approximation to the signal y(n) will be better. Therefore, the lower values of ε(i) show the position of a maximum or minimum point of the waves of a signal.

If the y(j) signal is very similar to a straight line, the value of ε(i) error signal will be very low causing false positives when detecting maximum and minimum points in the waves of a signal. However, it is possible to differentiate a straight line from a wave by merely looking at the value of the opening coefficient of parabola a, as this value will be minimal when the parabola approaches a straight line. This quality of the algorithm allows us to quickly detect the level of the isoelectric line in cardiac signals since it is only a question of setting limits concerning to the a constant.

Once the error signal is obtained, it is necessary to process it, storing the y(n) values when ε(i) is close to zero and higher than the defined limit. By doing this, it is possible to detect the R point and the average value of the isoelectric line.

### 3.2. Application of the Algorithm in the ECG Signal

As described above, the first stage was to choose the size of the window based on the sampling frequency, which in our case were signals sampled at a frequency of 500 Hz. Using Equation (2), we obtained *w* = 15, and therefore, the window had a length of 31 samples. Once the value of the window was found the algorithm was computed. The process can be graphically observed as the parabola $y^{\prime }(j)$ is adjusted by making a path along the signal y(n), which is an ECG signal. The parabola adapts to the waveform present in this window (Figure 6). In each iteration of *i*, the value of ε(i) is generated, obtained through Equation (9) (See Figure 7).

It is necessary to search the point where the error signal ε(i) is close to zero to detect the peaks of the R wave; this is achieved by storing the two previous samples of each iteration. A variable c is created to store the actual value of ε(i), and then, its two previous samples *c*_1_ and *c*_2_ are stored:(9)$$\begin{array}{l} c=\varepsilon [i]\\ c_{1}=\varepsilon [i-1]\\ c_{2}=\varepsilon [i-2] \end{array}$$

After, it is necessary to find the limit value of the opening coefficient of the parabola for exclusively detect the peak *R* with the width of the window previously calculated. This constant was obtained by visual inspection when calculating the opening coefficient (limit is equal to 5 × 10^−4^). The conditions for detecting peak *R* are as follows:(10)$$\begin{array}{l} a>5\times 10^{-4}\\ c_{2}>c_{1}<c \end{array}$$

With this, it is possible to determine the location of the *R* peaks in the signal y(n), as shown in Figure 8.

To detect the level of the isoelectric line, the limit imposed on the opening coefficient of the parabola must be changed, because it is necessary to make minimum this value to fit our parabola at a straight line. By visual inspection, we can choose the new limit as 1 × 10^−4^. Then, we search for the values of the signal samples y(j) when the signal ε(i) is close to zero. The conditions to obtain an approximate value of the isoelectric line are:(11)$$\begin{array}{l} a>1\times 10^{-4}\\ c<0.01 \end{array}$$

Thus, it is possible to obtain a constant value approximate to the voltage level of the isoelectric line, with the aim of being able to observe the polarity of the waves concerning to the line, as shown in Figure 9.

Once the peaks of the *R*-wave are detected, it is possible to calculate the average distance among all R points. With this value, we can build a window to segment the ECG signal in periods. In the signal shown in Figure 9, a total of 25 *R*-waves were detected. The average of samples among R points was equal to 384.28 samples. Since the sampling frequency was equal to 500 Hz, each sample had a duration of 2ms; thus, the average duration was 768.56 ms. The ECG signal was segmented using a window of 385 samples, simply moving the y(n) index forward and back half of the average sample, which would be 192 samples. This operation was performed the same number of times as the amount of detected waves minus two, because the first and last waves served as a reference framework for the beginning and the end of the segmentation. We stored the resulting signals in a matrix, whose dimension was the number of samples in the window by the sum of detected signals minus two. This matrix contained the signals separated by periods, which in the case of this example were 25 signals extracted from Figure 9 and represented individually in Figure 10. The separation of these signals was the previous stage to simplify the process of analyzing and detecting the rest of the peaks (P, Q, R, S, and T).

## 4. Detection of Peaks P, Q, R, S and T in Segmented ECG Signal

Later, we adjusted the algorithm to detect the rest of the peaks present in the segments of the ECG signal, for it reduced the size of the window to the duration of the smallest wave: the Q wave, which is approximately 20 to 30 ms [13,14]. To calculate the new window, we use:(12)$$w=0.012\times f_{s}$$

Giving as a result that *w* = 6; therefore, our window will be of 13 samples of length. Then, the algorithm was computed again, establishing a new limit with respect to the amplitude of the isoelectric line. These limits can be adjusted depending on how much noise remains in the signal. We looked for an upper limit whose value was slightly less than the amplitude of the smallest positive wave and a lower limit slightly higher than the amplitude of the smallest negative wave. In this way, we only detected the maximum points of the most outstanding waves of the segment. The last condition to detect all the peaks of the ECG signal was that there was a minimum distance between the detection of the peaks, because sometimes the signals still presented some kind of noise [19]. Significant fiducial points were detected in the segments previously stored in the matrix as shown in Figure 11. In this figure, we can clearly identify the P, Q, R, S and T peaks, which could be used in machine learning classifiers to discriminate normal and abnormal heartbeat patterns. 

In this sense, it was possible to differentiate between each peak by simply referencing the R point and conditioning the distances to both sides from this point, since each new point detected was the vertex of a parabola. Information about the amplitude of the wave and the width of its lobe is stored in the variables a and v.

## 5. Training, Test and Validation of the Algorithm

After heuristically adjusting the suitable analysis window for the detection of P, Q, R, S, and T peaks and establishing the appropriate thresholds, we trained and tested our algorithm using ECG signal records extracted from the PhysioNet ECG-ID database [19,20]. This database contains recordings of 310 ECG signals from 90 subjects. Each recording contains the I-lead ECG waveform, recorded during 20 s at a frequency of 500 Hz with a resolution of 12-bit over a nominal voltage range of ±10mV. For the training and tuning stage of the proposed approach, we randomly selected ECG segments with a duration of 5 s from 50 ECG records. Later, the trained algorithm was tested on the remaining 260 ECG signals, each 20 s long. 

Next, we performed a cyclic process to analyze each signal verifying that the maximum and minimum points found by the algorithm corresponded to the location of the P, Q, R, S, and T peaks. Detection process was assessed by calculating the temporal distance between detected points and reference points marked from the original signal. The maximal value to correctly identify a fiducial point was fixed with a threshold of 10 ms. Accuracy of correct detection over the testing set for each ECG peak was estimated by Sensitivity, which was calculated as the percentage of ECG records where the assessed distance (*d*) was below the preset threshold (*Thr*) when combination of detected (*T*_1_) and reference (T_2_) points were compared [22].
$$Sensitivity=100\frac{\sum_{i=1}^{N}d\left(ECGpeak_{i}\left(T_{2}\right),ECGpeak_{i}\left(T_{1}\right)\right)<Thr}{N}$$
where *N* is the number of tested ECG records.

Finally, to validate robustness of the proposed approach, we developed an ECG sensor to register and transmit ECG information to a mobile device [3,23,24]. A visual representation of the acquired signals in real time was based on implementation of our detection algorithm. 

## 6. Development of ECG Sensor

The sensor design included acquisition, coupling, amplification, filtering, digitalization, and transmission of ECG signals. The ECG signal captured by three identical electrodes, located in defined positions, was provided to low frequency amplifiers that perform three basic tasks: obtaining a differential signal, amplifying, and filtering it. These blocks were interconnected by impedance coupling modules that reduced the noise between circuits. The signal was filtered through low-pass and high-pass filters to improve the signal/noise ratio. The processed signal was digitized with an analog-to-digital converter (ADC) and transmitted by a Bluetooth module; both devices were embedded in an MCU. A 9V primary lithium battery with 1200 mAh capacity powered the ECG sensor.

To acquire the ECG signal, two electrodes were attached to the left and right arm and the reference electrode was placed far away from these on the right leg. The reference electrode plays the role of driving the user’s body to attenuate the common mode noise caused by external electromagnetic interference [25]. The analog input signal from two lead electrodes was initially amplified through an AD620 differential instrumentation amplifier (low cost and power) [26]. Before the next processing block, we coupled the impedance using a TL082 operational amplifier, configured as a voltage follower [27]. A low power OP97E operational amplifier was used to establish a user protection isolation, which protected the user from statics charges and suppresses voltages transients [28]. In addition, we included a processing block located after the voltage follower processes the ECG signal contaminated with noise and other disturbances produced by the acquisition system itself (Figure 11).

The function of this block was to deliver a clean signal and minimize disturbances by applying a non-inverting active amplifier and a band-pass filter, based on low-pass and high-pass filter. Both filters were designed using a Butterworth type approach, with a cut-off frequency of 0.5 Hz for the low part and 40 Hz for the high part, the conjunction of both filters allows removing unwanted frequency components from the ECG signal [16,29]. Later, a non-inverter adder added an appropriate carrier signal to obtain only positive voltage values. All these elements were designed using an LM324 low power quadruple operational amplifier [30] encapsulated in the processing block (Figure 12). Finally, the ECG output signal was ready to be read by any MCU using an ADC.

The ReadBEAR LABs BlendMicro that combined the Atmel ATmega32U4 microcontroller with a BLE module (Nordic nRF8001, Nordic Semiconductor, Trondheim, Noruega) [31,32] was used as an MCU to process the signal through its built-in ADC and transmit the information via Bluetooth. The Generic Access Profile (GAP) layer of the BLE was the unit that controlled connections and advertising to determine how two devices interacted with each other by assigning roles. The ECG sensor (Figure 13), which includes the low-level processing module and the BlendMicro, communicated wirelessly with a Bluetooth receiver connected to a computer, where the data was received and stored so that in a next step it can be visually represented by applying the algorithm proposed.

## 7. Results

During the training stage, the detection algorithm performed with a Sensitivity of 100% to identify the P, Q, R, S, and T points for ECG signals of the training set. For the stage of test, R, S, P, and T peaks were detected with a successful rate of 100%. Only detection of the Q-peak showed a slightly lower Sensitivity (97.5%), mainly because in some cases its amplitude was confused with the isoelectric line. Total accuracy in terms of Sensitivity for detection of fiducial points in testing set is shown in Table 1. 

In Figure 14, we show examples for correct and erroneous detection of peaks from ECG segment. Figure 14A indicates true estimation for all fiducial points; however, sometimes the algorithm failed to detect the Q-point (Figure 14B). This error was caused by the closeness between the Q-estimation and isoelectric line voltage in 2.5% of the cases. For 7020 Q-points analyzed, only 175 not were detected.

With the ECG sensor developed, we acquired raw ECG signals in order to validate the robustness of our approach for segmenting the signals and correctly identifying their fiducial points. In Figure 15, we show the ECG signal processing in real time: (A) record of the original signal; (B) signal filtering; (C) deployment of the unstable isoelectric line; and (D) stable isoelectric line after subtracting the polynomial function.

In Figure 16, we show the segmentation process divided in three steps: (A) approximation of the isoelectric line; (B) detection of the R points as reference point to identify the rest of fiducial points; (C) detection of the P, Q, S, and T points for three different ECG segments.

## 8. Conclusions

Most of the processing algorithms for ECG signals have analyzed the accurate detection of the QRS complex, performing with a success rate close to 100%. However, aspects such as processing speed and efficient detection of all ECG peaks have not been sufficiently addressed. Conversely, we proposed a fast and straightforward method based on an algorithm of low computational cost for segmentation of cardiac signals and correct detection of their fiducial points. 

This method provides storing information from parabolic wave functions from the shape of their vertex, with the goal of improving its interpretation. Thus, the algorithm allows segmenting an ECG signal in periods and identifying the P, Q, R, S, and T points, which can provide relevant information to discriminate normal and abnormal heartbeat patterns. Because through the analysis of these ECG peaks, automated approaches such as machine learning classifiers can contribute to diagnose cardiac pathologies with higher accuracy. 

In this sense, our algorithm implemented a simple and eager linear regression process to analyze 260 ECG signals, obtaining an average Sensitivity of 99.5% to identify all P, Q, R, S, and T peaks. Furthermore, we validated the robustness of the proposed approach by developing an ECG sensor to acquire raw ECG signals in real time, whose visual representation was based on implementation of our detection algorithm.

Although the average Sensitivity of our approach was high, there was an error rate around of 3% in the correct detection of point Q. On the other hand, in the dataset selected for the training and testing stages there were few examples of non-sinusal or abnormal beats, which can limit the generalization capacity of the detection algorithm. Therefore, our approach requires the analysis of more ECG data with cardiac disorders. Thus, with further validation, the proposed method can be part of a cost-effective strategy for primary diagnosis of potential arrhythmias to improve preventive health care.

## Figures and Tables

**Figure 1 sensors-19-00775-f001:**
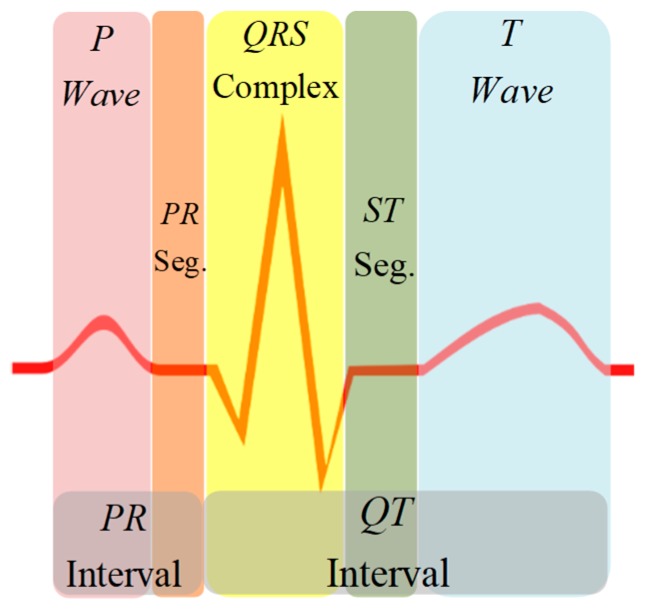
Electrocardiogram (ECG) Signal Morphology.

**Figure 2 sensors-19-00775-f002:**
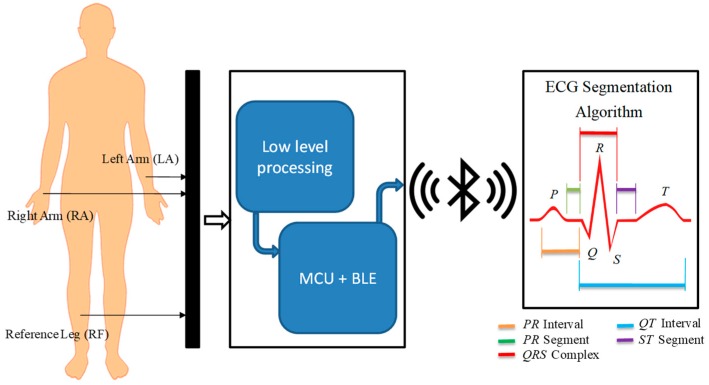
General architecture of ECG sensor and processing system.

**Figure 3 sensors-19-00775-f003:**
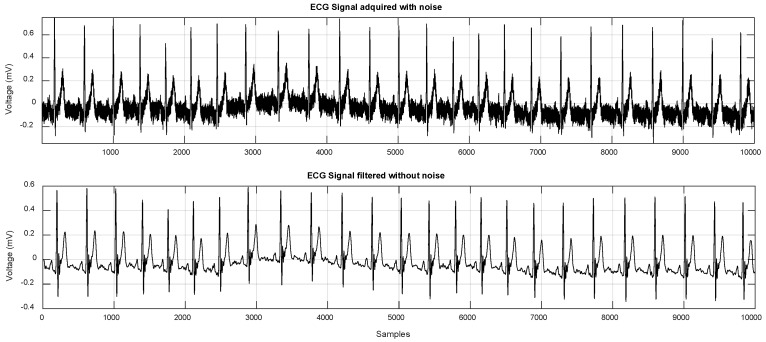
Comparison between the ECG signal with noise (**upper**) and the filtered ECG signal (**lower**).

**Figure 4 sensors-19-00775-f004:**
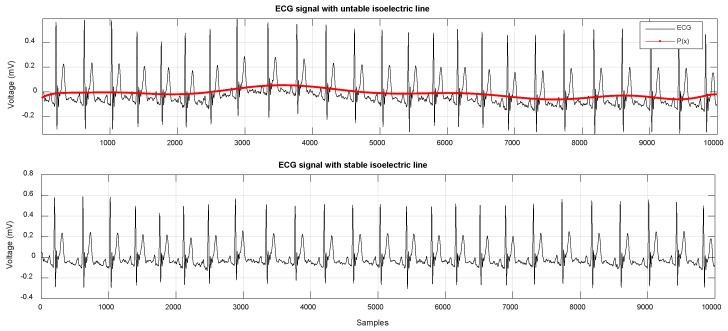
ECG signal showing changes in the isoelectric line (**upper**) and ECG signal obtained after subtraction of the polynomial function (**lower**).

**Figure 5 sensors-19-00775-f005:**
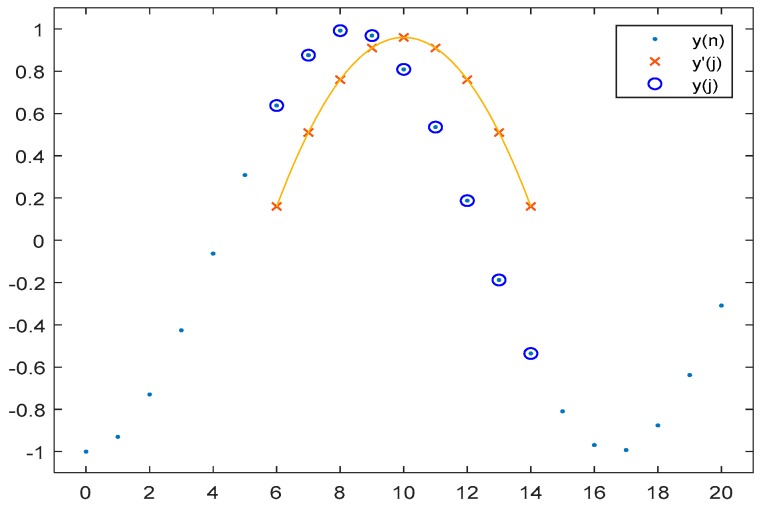
Parabola $y^{\prime }(j)$ approximating to the ECG signal within the window.

**Figure 6 sensors-19-00775-f006:**
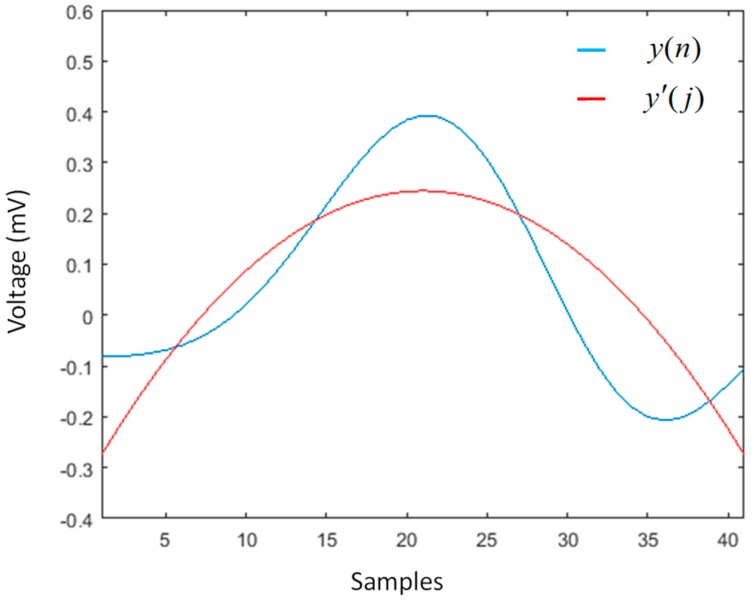
Parabola $y^{\prime }(j)$ approximating to the ECG signal within the window.

**Figure 7 sensors-19-00775-f007:**
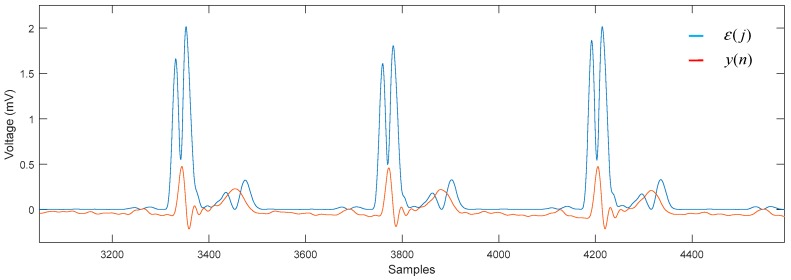
Comparison of the error signal against the ECG signal.

**Figure 8 sensors-19-00775-f008:**
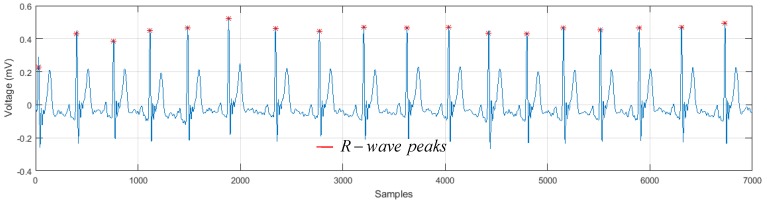
Detection of *R*-wave peaks.

**Figure 9 sensors-19-00775-f009:**
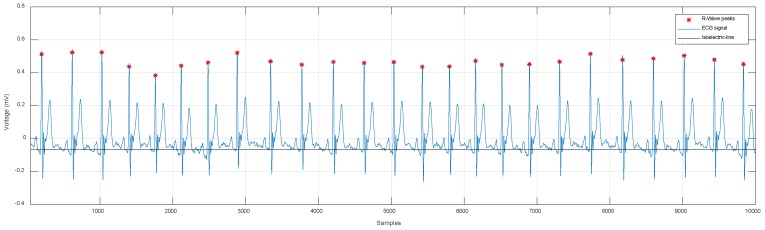
Approximated isoelectric line.

**Figure 10 sensors-19-00775-f010:**
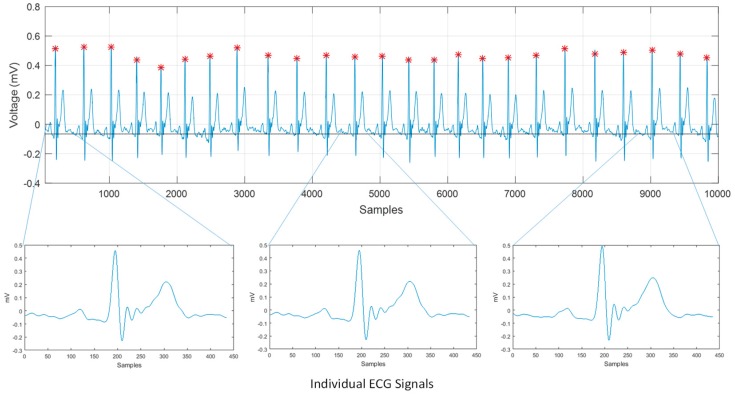
ECG signal separated in periods of automatic form applying the proposed algorithm.

**Figure 11 sensors-19-00775-f011:**
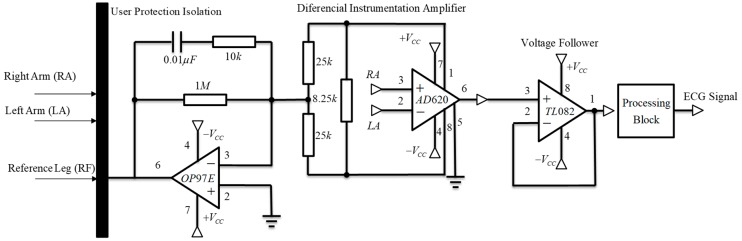
Schematic representation of ECG amplifier and isolation circuit.

**Figure 12 sensors-19-00775-f012:**
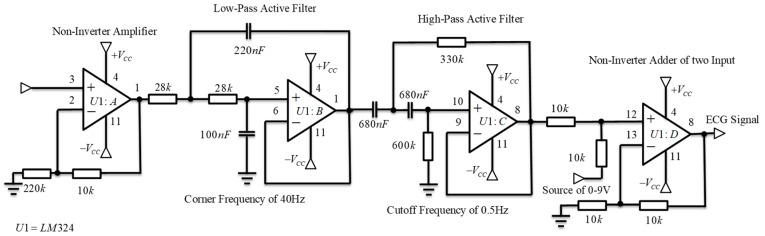
Encapsulation of processing block with amplifiers and active-filter circuits.

**Figure 13 sensors-19-00775-f013:**
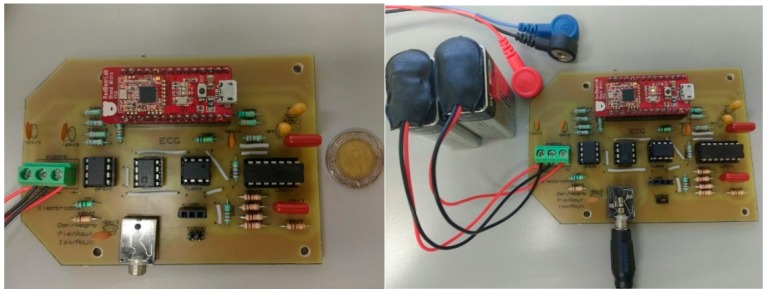
ECG Sensor that include the low-level processing (amplifier, filter and isolation circuit) and top level processing (analog-to-digital converter, ADC; microcontroller unit, MCU; Bluetooth Low Energy, BLE).

**Figure 14 sensors-19-00775-f014:**
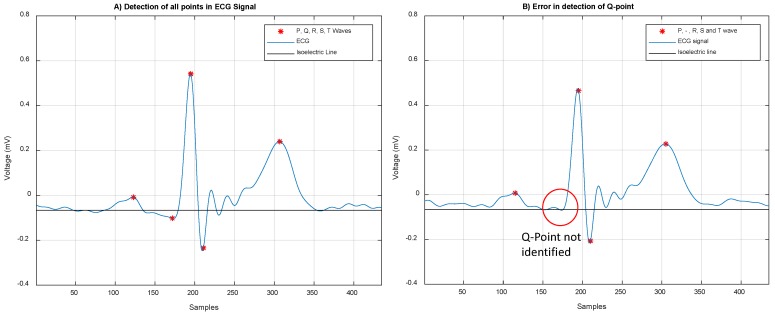
(**A**) Correct detection of P, Q, R, S, and T points from ECG signal and (**B**) erroneous detection of Q point when its value is close to the isoelectric line voltage.

**Figure 15 sensors-19-00775-f015:**
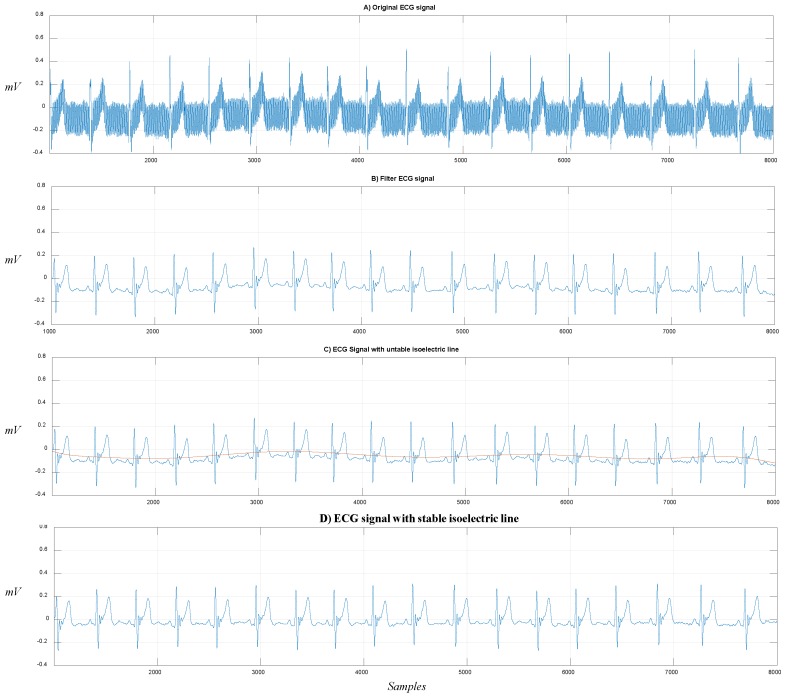
ECG signal processing: (**A**) original signal, (**B**) filtered signal, (**C**) unstable isoelectric line, (**D**) ECG signal ready to be processed and segmented.

**Figure 16 sensors-19-00775-f016:**
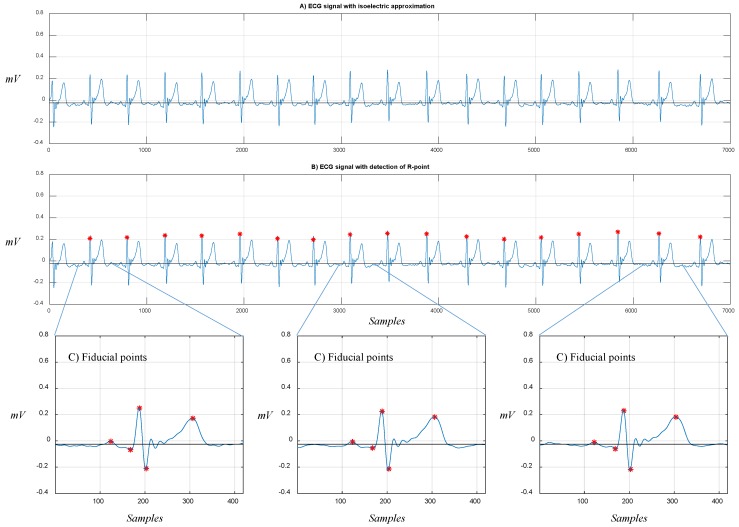
ECG signal processing: (**A**) approximation of the signal to the isoelectric line, (**B**) detection of the R points along the signal, (**C**) detection of the P, Q, R, S, and T (fiducial) points.

**Table 1 sensors-19-00775-t001:** Sensitivity for detection of the P, Q, R, S, and T peaks from 260 ECG signal assessed on testing stage.

R-Peak	Q-Peak	S-Peak	P-Peak	T-Peak
100%	97.5%	100%	100%	100%

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
