# Peer review of "Segmentation of the ECG Signal by Means of a Linear Regression Algorithm"

_sensors, 2019, doi:10.3390/s19040775_

Round 1
Reviewer 1 Report
The field of waves detection over the ECG is a mature field, with success percentages close to 100%. Nevertheless the authors points that one one of the advantages of the method proposed in this paper is its simplicity. Unfortunately the authors have not done any experimental verification of the computational cost.
According to table I the algorithm is "perfect", with a 100 % of success. The problem is that the validation is very poor. No separate training and validation sets are used. No information is given about how a wave is supposed to be correctly detected (how many samples of error are permitted between reference annotation and experimental one?).
Minor comments:
- Line 147: m>N correct?
- Equation 3: y'(j) will be better
- Equation 5: y'(x)
- Fig 10 could be removes, is repeated on figure 11.
- Fig 11: caption is in Spanish
- Fig 14: is not referenced in the text and caption is not correct
- Please review references format, for instance 6 and 10 have not the same format
- Reference 29 and 30 are not correctly ordered.
Author Response
RESPONSE TO REVIEW
REVIEWER 1:
The field of waves detection over the ECG is a mature field, with success percentages close to 100%. Nevertheless the authors points that one one of the advantages of the method proposed in this paper is its simplicity. Unfortunately the authors have not done any experimental verification of the computational cost.
Response: We have added in the conclusions an improvement that will be made in future work, which consists of implementing the algorithm presented in this document in an embedded device in order to obtain precise data of computational load, number of operations carried out, response time of the complete operation of the algorithm, among others..
According to table I the algorithm is "perfect", with a 100 % of success. The problem is that the validation is very poor. No separate training and validation sets are used. No information is given about how a wave is supposed to be correctly detected (how many samples of error are permitted between reference annotation and experimental one?).
Response: In section 5 of the results, we have added detailed information about the algorithm training process and the verification test. With this additional information that is detailed in the document, it is reported that a small sample of the database was used to find the best parameters of the algorithm and that in the validation stage the rest of the records in the database were used. Regarding the second question, the algorithm detects the waves through an unsupervised algorithm, in this case, it is not compared with possible targets but it may be an option to be analyzed in future work.
Minor comments:
- Line 147: m>N correct?
Response: the equation was modified in section 3 to avoid confusion.
- Equation 3: y'(j) will be better
Response: the suggestions of the reviewers were addressed in section 3.
- Equation 5: y'(x)
Response: the suggestions of the reviewers were addressed in section 3
- Fig 10 could be removes, is repeated on figure 11.
Response: the suggestions of the reviewers were taken care of when eliminating figure 10 and the necessary changes were made in the text of sections 4 and 5.
- Fig 11: caption is in Spanish
Response: the suggestions of the reviewers in the results section were attended.
- Fig 14: is not referenced in the text and caption is not correct
Response: the suggestions of the reviewers in the results section were attended.
- Please review references format, for instance 6 and 10 have not the same format
Response: the suggestions of the reviewers in the references section were attended.
- Reference 29 and 30 are not correctly ordered.
Response: the suggestions of the reviewers in the references section were attended.

Reviewer 2 Report
The examples provided by the authors are straightforward. Whereas this can be acceptable for illustration purposes, some difficult segments of ECG should also be included. Comparison with other algorithms and with usually used databases in the field should be made.
Author Response
RESPONSE TO REVIEW
REVIEWER 2:
The examples provided by the authors are straightforward. Whereas this can be acceptable for illustration purposes, some difficult segments of ECG should also be included. Comparison with other algorithms and with usually used databases in the field should be made.
Response: In this first stage of the segmentation project, it was possible to obtain and detect the important points that give rise to the internal waves of an ECG signal. However, in the validation part, complex waves were not considered because many of them are related to certain pathologies and we intend in a next step, to use the collected information to use it in some recognition algorithm to establish classes and detect pathologies. Regarding the second suggestion, a comparison was made with a work that was mentioned in the introduction where they detect the QRS complex, and in the conclusions, it is mentioned that the proposed algorithm manages to detect the QRS complex and also, the P and T waves

Reviewer 3 Report
The manuscript presents a methodology for detection of fiducial points in the electrocardiogram. Although there are many published methods for QRS detection, the one used in this study sounds interesting. Moreover, the authors claim that it is also applicable for identification of P, Q, S, T waves, which is beneficial.
I have the following critical remarks, questions and recommendations:
1) The language needs considerable improvement. Intentionally I will not provide any examples for sentences that need to be corrected, since the authors should not focus on particular sentences but should considerably improve all the text. Even if the authors fulfill all other requirements, if the language is not corrected the manuscript will not be suitable for publication.
2) The used term “critical points in the signal” is not correct. Usually, “fiducial points” is used. Change it everywhere!
3) Section Introduction:
- The authors have written: “These data include the position and magnitude of the PR and QT intervals, and the PR and ST segments.” Normally, when we address the intervals in the ECG, we consider their duration. The magnitude (amplitude) is measured for the waves (P, QRS, T).
- The last paragraph (starting with “”) is not appropriate for this section. Instead, at the end of the Introduction the authors should highlight the aim of the study.
- The following sentence is not clear. Rewrite it! “The third stage consists of using the algorithm in the signals obtained by segmenting the ECG signal, to detect the maximum points of the remaining waves to store the information obtained from each of the waves P, Q, R, S, T. All this is done using the mathematical software tool MATLAB [18].”
4) Section “ECG signal processing”:
- The power-line interference could be also with 50 Hz.
- I could not understand the sense of this sentence: “The application of a Butterworth low-pass filter presents a smooth transition between the passband and the cut band.”
- “A low-pass filter with a 100Hz cutoff frequency was applied to eliminate high frequency and small amplitude variations.” – In fact, the filter is applied to suppress all high-frequency noises but not only these with small amplitudes.
- ‘a’ and ‘b’ are filter coefficients (not filter parameters).
- “the deviation present in the baseline” – the accepted term is “baseline wander”
5) Section “Detection of R-point and isoelectric line”:
- “m>N” – check this, since it means that the analysed samples are more than the recorded.
- The following sentence is not clear: “We define i as the index located in the center of the window which runs through the signal sample by sample. Inside that window, the signal fragment y(n) , now known as y( j) , approximates to a parabolic function y’( j) where j is the new index that comprises the signal samples included inside the w+1 window size.”
6) Section “Detection of waves P, Q, R, S, T in ECG signal segmented” (rename it to “Detection of waves P, Q, R, S, T in segmented ECG signal”):
- “for it reduces the size of the window to the duration of the smallest wave” – this is not clear. Stop the sentence before this statement and write the next sentence with clear subject.
7) Up to this point of the manuscript (the description of the method is finished) it is not clear how are the threshold values of the algorithm adjusted. Training and test databases should be defined (e.g. in section “ECG databases”, which should precede the description of the methodology). All adjustments should be done on the training database. The test database should be used only for testing! The database description (presented in section Results) should be moved and extended. Are there any annotations of P, Q, R, S, T waves in it?
8) Section Results:
- In what time window around the annotation (if present in the database) the wave detection is considered as correct? Details about the test procedure should be presented.
- Table 1 – the caption is “Percentage of success in the detection of the P, Q, R, S, and T”. Have you done detection of the Q and S waves? If yes – separate the results in column ‘QRS’ into ‘Q-point’ and ‘S-point’. If no, what is the difference between the detection of ‘R-point’ and ‘QRS’?
- Normally, when a ‘wave detector’ is trained and tested the following statistical indices should be declared:
i) Sensitivity Se = TP/(TP+FN)
ii) Positive predictive value PPV = TP/(TP+FP),
where TP is true positive detections (number of correctly detected waves), FN is false negative detections (number of waves that are erroneously not detected), FP is false positive (number of erroneously detected waves).
- Provide separate results for training and testing. Provide also the number of ECG recordings and the number of tested beats for which you declare the accuracy results (Se, PPV).
- Change the language of the caption of Figure 11!
- Figure 14 is not addressed in the text. The caption is part of the caption of figure 11.
Author Response
RESPONSE TO REVIEW
REVIEWER 3:
The manuscript presents a methodology for detection of fiducial points in the electrocardiogram. Although there are many published methods for QRS detection, the one used in this study sounds interesting. Moreover, the authors claim that it is also applicable for identification of P, Q, S, T waves, which is beneficial.
I have the following critical remarks, questions and recommendations:
1) The language needs considerable improvement. Intentionally I will not provide any examples for sentences that need to be corrected, since the authors should not focus on particular sentences but should considerably improve all the text. Even if the authors fulfill all other requirements, if the language is not corrected the manuscript will not be suitable for publication.
Response: the suggestions of the reviewers were taken care of when revising the text and improving it.
2) The used term “critical points in the signal” is not correct. Usually, “fiducial points” is used. Change it everywhere!
Response: the suggestion of the reviewer was attended and the terminology was changed to be in line with the line of work.
3) Section Introduction:
- The authors have written: “These data include the position and magnitude of the PR and QT intervals, and the PR and ST segments.” Normally, when we address the intervals in the ECG, we consider their duration. The magnitude (amplitude) is measured for the waves (P, QRS, T).
Response: the suggestion of the reviewer was attended and the text was corrected.
- The last paragraph (starting with “”) is not appropriate for this section. Instead, at the end of the Introduction the authors should highlight the aim of the study.
Response: the suggestion of the reviewer was attended and the text was corrected.
- The following sentence is not clear. Rewrite it! “The third stage consists of using the algorithm in the signals obtained by segmenting the ECG signal, to detect the maximum points of the remaining waves to store the information obtained from each of the waves P, Q, R, S, T. All this is done using the mathematical software tool MATLAB [18].”
Response: the suggestion of the reviewer was attended and the text was corrected “The third stage consists of applying the algorithm again throughout the ECG signal to detect the maximum and minimum points within an analysis window, which generate information on the location of the P, Q, R, S, T waves. We performed the mathematical analysis using the MATLAB [18] (mathematical software tool), which allowed to obtain the coordinates of the maximum and minimum points corresponding to the location of the ECG signal waves”
4) Section “ECG signal processing”:
- The power-line interference could be also with 50 Hz.
Response: the suggestion of the reviewer was attended and the text was corrected.
- I could not understand the sense of this sentence: “The application of a Butterworth low-pass filter presents a smooth transition between the passband and the cut band.”
Response: The use of basic or ideal filters on multi-frequency signals means that important information present in the signal is suppressed. The use of butterworth filters allows the transition between the cutoff frequency and the passband is not severe and allows to pass some frequencies that are above the cutoff frequency and contain relevant information of the signal. This will affect the processed signal less. Considering this, the text was corrected to better explain the sentence: “The application of a Butterworth low-pass filter presents a smooth transition between the frequencies belong to passband and the frequencies that are over cut band. This allows essential ECG signal information to be kept present during a smooth transition”.
- “A low-pass filter with a 100Hz cutoff frequency was applied to eliminate high frequency and small amplitude variations.” – In fact, the filter is applied to suppress all high-frequency noises but not only these with small amplitudes.
Response: the suggestion of the reviewer was followed and the text was modified to avoid confusion
- ‘a’ and ‘b’ are filter coefficients (not filter parameters).
Response: the suggestion of the reviewer was followed and the text was modified to avoid confusion
- “the deviation present in the baseline” – the accepted term is “baseline wander”
Response: the suggestion of the reviewer was followed and the text was modified to avoid confusion
5) Section “Detection of R-point and isoelectric line”:
- “m>N” – check this, since it means that the analysed samples are more than the recorded.
Response: the suggestion of the reviewer was followed and an adaptation was made to the whole section 3.1 to better explain the algorithm and avoid confusion.
- The following sentence is not clear: “We define i as the index located in the center of the window which runs through the signal sample by sample. Inside that window, the signal fragment y(n) , now known as y( j) , approximates to a parabolic function y’( j) where j is the new index that comprises the signal samples included inside the w+1 window size.”
Response: the suggestion of the reviewer was followed and an adaptation was made to the whole section 3.1 to better explain the algorithm and avoid confusion.
6) Section “Detection of waves P, Q, R, S, T in ECG signal segmented” (rename it to “Detection of waves P, Q, R, S, T in segmented ECG signal”):
Response: the suggestion of the reviewer was attended and the text was adapted.
- “for it reduces the size of the window to the duration of the smallest wave” – this is not clear. Stop the sentence before this statement and write the next sentence with clear subject.
Response: the suggestion of the reviewer was attended and the text was adapted.
7) Up to this point of the manuscript (the description of the method is finished) it is not clear how are the threshold values of the algorithm adjusted. Training and test databases should be defined (e.g. in section “ECG databases”, which should precede the description of the methodology). All adjustments should be done on the training database. The test database should be used only for testing! The database description (presented in section Results) should be moved and extended. Are there any annotations of P, Q, R, S, T waves in it?
Response: The suggestion of the reviewer was attended and the text was adapted to explain the training process and the validation process, this adaptation was carried out in the section 4 and 5. “After having heuristically adjusted the suitable analysis window for the detection of P, Q, R, S, and T waves, and established the appropriate thresholds, the algorithm began training using ECG signal records extracted from the PhysioNET database [20]. We used a subset of records from the ECG signal bank for the training and tuning of the algorithm, computed by 50 signals with a duration of 5 seconds. Next, we performed a cyclic process to analyze each of these signals verifying that the maximum and minimum points found by the algorithm corresponded to the location of the P, Q, R, S, and T waves. Finally, this series of steps gave the guideline to start the final tests of the algorithm with new registers and thus verify the robustness of the system proposed in the next section”.
8) Section Results:
- In what time window around the annotation (if present in the database) the wave detection is considered as correct? Details about the test procedure should be presented.
Response: The suggestion of the reviewer was attended and the text was adapted to explain what was requested.
- Table 1 – the caption is “Percentage of success in the detection of the P, Q, R, S, and T”. Have you done detection of the Q and S waves? If yes – separate the results in column ‘QRS’ into ‘Q-point’ and ‘S-point’. If no, what is the difference between the detection of ‘R-point’ and ‘QRS’?
Response: The suggestion of the reviewer was attended and, table 1 and the text were adapted to explain the requested.
- Normally, when a ‘wave detector’ is trained and tested the following statistical indices should be declared:
i) Sensitivity Se = TP/(TP+FN)
ii) Positive predictive value PPV = TP/(TP+FP),
where TP is true positive detections (number of correctly detected waves), FN is false negative detections (number of waves that are erroneously not detected), FP is false positive (number of erroneously detected waves).
Response: The suggestion of the reviewer was attended and it is justified how the percentages in the detections were obtained. In the case of the proposed algorithm, if the analysis window were wider if false positives or maximum points were obtained where there should not be, this is why during training the window size and algorithm parameters were adjusted so that there is a need to add these statistical indices.
- Provide separate results for training and testing. Provide also the number of ECG recordings and the number of tested beats for which you declare the accuracy results (Se, PPV).
Response: The recommendation of the reviewer was followed and the text was modified to explain the results obtained in the training stage and those corresponding to the validation stage.
- Change the language of the caption of Figure 11!
Response: the reviewer's suggestion was taken care of and the text was modified.
- Figure 14 is not addressed in the text. The caption is part of the caption of figure 11.
Response: the reviewer's suggestion was taken care of and the text was modified.

Round 2
Reviewer 1 Report
The text has been improved after this round.
I have only doubts about the validation process. When you mark a beat as correctly detected I suppose that is because the difference between reference annotation and your annotation is below a fixed threshold (because perfect coincidence is impossible). Then, what threshold have you used? Some tools for beat detectors validation like bxb form Physionet used a default threshold of 0.15 seconds (the one proposed in ANSI/AAMI standard).
Unfortunately most part of papers I have read in this field do not explain this point, so it could be acceptable for this paper.
Author Response
The text has been improved after this round.
I have only doubts about the validation process. When you mark a beat as correctly detected I suppose that is because the difference between reference annotation and your annotation is below a fixed threshold (because perfect coincidence is impossible). Then, what threshold have you used? Some tools for beat detectors validation like bxb form Physionet used a default threshold of 0.15 seconds (the one proposed in ANSI/AAMI standard).
Unfortunately most part of papers I have read in this field do not explain this point, so it could be acceptable for this paper.
We thank the reviewer for the thoughtful comment. To help clarify this topic, we included information about the validation process in Section 5 (Line 294):
“Detection process was assessed calculating the temporal distance between detected points and reference points marked from the original signal. The maximal value to correctly identify a fiducial point was fixed with a threshold of 10ms.”

Reviewer 2 Report
Authors have given response to the suggestions. Still, there are little representation of other non-sinusal beats, like ventricular ones. This can be a limitation in the experimental setup.
Author Response
Authors have given response to the suggestions. Still, there are little representation of other non-sinusal beats, like ventricular ones. This can be a limitation in the experimental setup.
This is an interesting remark. Therefore, we added the following statement in our Conclusions (Line 410):
“On the other hand, in the dataset selected for the training and testing stages there are few examples of non-sinusal or abnormal beats, which can limit the generalization capacity of the detection algorithm. Therefore, our approach requires identification of ECG data for cardiac disorders.”

Reviewer 3 Report
The revised version of the manuscript corresponds to some of the remarks in my previous review. However, I will repeat the remarks, which are not adequately addressed and that prevent me from a positive decision for the manuscript.
1) The authors claim that the language is improved but I am not convinces with this. Obviously, they need third party help in this process.
2) Section Introduction:
- The last paragraph (starting with “In this paper, an algorithm is presented based on a method …”) is not appropriate for this section. Instead, at the end of the Introduction the authors should highlight the aim of the study. – Not addressed in the revised version. The aim of the study is also not presented at the end of section Introduction.
3) Section “ECG signal processing”:
- Substitute “interference from the f = 50-60 Hz power-line” with “power-line interference (with frequency 50 Hz or 60 Hz)”
4) I could not find the answer related to my question: Are there any annotations of P, Q, R, S, T waves in it? – If ‘NO’ – how did you perform the training and testing? Explain! Did you judje the accuracy of detection via visual observation? If ‘YES’ – what is the maximal distance between detected point and annotated point that you consider as acceptable (i.e. consider the detection as correct).
5) Section Results:
- Normally, when a ‘wave detector’ is trained and tested the following statistical indices should be declared:
i) Sensitivity Se = TP/(TP+FN)
ii) Positive predictive value PPV = TP/(TP+FP),
where TP is true positive detections (number of correctly detected waves), FN is false negative detections (number of waves that are erroneously not detected), FP is false positive (number of erroneously detected waves).
- Provide separate results for training and testing. Provide also the number of ECG recordings and the number of tested beats for which you declare the accuracy results (Se, PPV).
Author Response
The revised version of the manuscript corresponds to some of the remarks in my previous review. However, I will repeat the remarks, which are not adequately addressed and that prevent me from a positive decision for the manuscript.
1) The authors claim that the language is improved but I am not convinces with this. Obviously, they need third party help in this process.
We have gone through the manuscript and improved the grammar.
2) Section Introduction:
- The last paragraph (starting with “In this paper, an algorithm is presented based on a method …”) is not appropriate for this section. Instead, at the end of the Introduction the authors should highlight the aim of the study. – Not addressed in the revised version. The aim of the study is also not presented at the end of section Introduction.
We agree with the point of view of the reviewer. Therefore, we included the following statement in the Introduction (Line 83):
“In this study, we propose a novel method for detecting fiducial points of ECG waves, using linear regression to identify maximums and minimums from an acquired ECG signal.”
3) Section “ECG signal processing”:
- Substitute “interference from the f = 50-60 Hz power-line” with “power-line interference (with frequency 50 Hz or 60 Hz)”
We substituted the statement in Line 111.
4) I could not find the answer related to my question: Are there any annotations of P, Q, R, S, T waves in it? – If ‘NO’ – how did you perform the training and testing? Explain! Did you judge the accuracy of detection via visual observation? If ‘YES’ – what is the maximal distance between detected point and annotated point that you consider as acceptable (i.e. consider the detection as correct).
We thank the reviewer for the thoughtful comment. To help clarify this issue, we included information about the validation process in Section 5 (Line 294):
“Detection process was assessed calculating the temporal distance between detected points and reference points marked from the original signal. The maximal value to correctly identify a fiducial point was fixed with a threshold of 10ms.”
5) Section Results:
- Normally, when a ‘wave detector’ is trained and tested the following statistical indices should be declared:
i) Sensitivity Se = TP/(TP+FN)
ii) Positive predictive value PPV = TP/(TP+FP),
where TP is true positive detections (number of correctly detected waves), FN is false negative detections (number of waves that are erroneously not detected), FP is false positive (number of erroneously detected waves).
- Provide separate results for training and testing. Provide also the number of ECG recordings and the number of tested beats for which you declare the accuracy results (Se, PPV).
We agree with the reviewer that clarification on all these accuracy results was needed.
Consequently, we added the following explanation in Line 296 (including the formula to calculate the Sensitivity):
“Accuracy of correct detection over the testing set for each ECG peak was estimated by Sensitivity, which was calculated as the percentage of ECG records where the assessed distance (d) was below the preset threshold (Thr) when combination of detected (T1) and reference (T2) points were compared, where N is the number of tested ECG records.”
In addition, we provided the number of tested and detected Q-points, which was the only ECG peak where the algorithm performed below 100% accuracy (Line 370):
“This error caused by the closeness between the Q-estimation and isoelectric line voltage was present in 2.5% of the cases. For 7020 Q-points analyzed, only 175 not were detected.”

Round 3
Reviewer 3 Report
The manuscript is suitable for publication in its present form.